# Determinants of wheat residue burning: Evidence from India

**Adrian A. Lopes, Dina Tasneem, Ajalavat Viriyavipart** *

Department of Economics, School of Business Administration, American University of Sharjah, Sharjah, United Arab Emirates

* aviriyavipart@aus.edu

## Abstract

Postharvest crop residue burning (CRB) is a major policy issue in several developing countries because of harmful environmental and public health consequences. While the literature has extensively examined the reasons for rice CRB, much less is known about wheat residue management. This paper explores the drivers of CRB of wheat in India and relates it to farmers' prior decisions related to CRB of rice in the preceding season. Using primary data on residue management practices of 301 Indian farmers, whom we tracked over two consecutive harvests of rice (2018) and wheat (2019), we find that farmers are significantly more likely to burn wheat residue if they had previously burned rice residue. The possibility of this linkage or spillover increases the likelihood of wheat residue burning by 15.6 to 21 percent. Furthermore, farmers are undertaking wheat CRB despite the positive net benefit of choosing non-burning alternatives to manage crop residue. Our results suggest that ensuring well-functioning markets for crop residue, awareness campaigns, and recognition of spillover effects of residue management across crops over time can enable policies to promote pro-environmental postharvest choices.

## Introduction and motivation

Crop residue burning (CRB) refers to the burning of crop residue, following the harvest season. CRB usually takes place in open fields and is used as a strategy by farmers to prepare farms for the next cropping cycle. CRB occurs with seasonal regularity across much of South and East Asia in countries such as India, the Philippines, China, and Thailand. The adverse implications of this practice have been recognized with increasing urgency in recent times. Of note is the substantial increase in air pollution that has significant implications for environmental and public health outcomes [1–4]. The polluted air spreads from its place of origin to neighboring rural and urban areas [5–10]. This form of air pollution has been associated with increases in respiratory illnesses and public health costs [11,12]. In addition to its contribution to air pollution, CRB has been known to negatively affect soil quality by reducing organic matter [13]. This in turn can result in lower farm yields over prolonged periods of time [14]. CRB therefore becomes pertinent from the perspectives of sustainable farmland management practices, the natural environment, as well as public health.

**Data Availability Statement:** All relevant data are within the manuscript and its Supporting Information files.

**Funding:** AL and AV received the Faculty Research Grant (#FRG17-R-49) from the American University of Sharjah. The funder had no role in

study design, data collection and analysis, decision to publish, or preparation of the manuscript.

**Competing interests:** The authors have declared that no competing interests exist.

In the case of India, much of the recent research on CRB has focused on the rice crop in particular [3,5,15–21]. Rice is the major summer crop in large parts of the Indo-Gangetic plains. Typically sown during June-July, it is harvested in November-December. Farmers typically do not reap rice residue and choose to burn it because of its limited usability. In contrast to rice, little or no attention has been paid to CRB practices related to wheat which constitutes the second, winter crop in large parts of Northern and Eastern India [21,22]. Similarly, attention has not focused on other parts of India where crops such as rice, sugarcane, and oilseed are also subject to postharvest residue burning; for example, in western and southern India in states such as Maharashtra, Karnataka, and Tamil Nadu [16,21,23]. Wheat is sown during December-January and harvested in April-May. Unlike rice residue that is burned, wheat residue is characteristically reaped post-harvest because of its tradable value as cattle fodder [24]. [25] report that forty-five percent of wheat residue in Punjab is used as cattle fodder. They estimate that 81 and 48 percent of farmers burn rice and wheat residue, repectively. Similarly [26], estimate these numbers to be 40 and 22 percent, repectively. Although wheat residue burning has not been as common as rice straw burning traditionally [27], report that farmers in northern India have increasingly been using CRB as a postharvest practice for wheat as well. A recent examination of satellite data on fire counts in northwest India suggests that wheat residue burning has been on the rise between 2014 and 2020 [22]. It is against this background that our study examines the socioeconomic reasons that drive the practice of CRB for wheat residue in the state of Haryana, India.

Using primary data on farmers' postharvest residue management practices this study contributes to an emerging evidence base of wheat residue burning in India. The curiosity that wheat CRB occurs at all, even though its residue retains value as cattle fodder, merits closer investigation. We do this firstly by conducting a benefit-cost test to examine whether it is in the farmer's financial interest to choose CRB instead of non-burning alternatives for wheat residue management. Secondly, we examine the extent to which a farmer's decision to burn wheat residue is conditional on their CRB decision of the (preceding) rice crop. We explore the possibility of a relationship between the burning of rice and wheat residues since the cultivation of both crops is intensively mechanized with combine harvesters, which is an important reason for choosing CRB.

To our knowledge, this is the first study to draw a link between CRB practices across crops grown in different agricultural seasons. Our study is well-placed to investigate these relationships for the following reasons. First, the agricultural practices in Haryana reflect the typical rice-wheat farming system. This allows us to identify the factors that can explain the reasons for wheat CRB. Second, our analysis relies on primary data of residue management practices collected over two consecutive harvest seasons of rice (November-December in 2018) and wheat (April-May in 2019). Furthermore, the findings from our previous study on the drivers of rice CRB in Haryana [17] indicated that the perception of CRB being a common practice significantly increases CRB, while the perception that it lowers soil quality makes CRB less likely. However, an awareness of its negative environmental effects did not lower its occurrence–suggesting a public goods characteristic of CRB. Given that this study examines the wheat CRB choices of a subset of the same rice farmers in a subsequent harvest season, it allows us to establish how the decision to burn wheat residue is related to burning rice residue in the previous season.

The rest of the paper is organized as follows. In the next section, we describe the background of CRB in India to provide context to our study. We follow this with a description of our sample selection and empirical strategy. In the Methodology section, we lay out the components for a benefit-cost test of wheat CRB and follow that with a description of a regression framework with a set of testable hypotheses. The Results section presents key findings. This is

followed by a discussion section with pertinent policy recommendations and a conclusion section.

## Background of CRB in India

The earliest documented cases of crop residue burning (CRB) in India were in the mid-1980s [28]. Farming in India began to be intensely mechanized in the 1970s from the introduction of combine harvesters [29]. The use of machinery spread across the north Indian states of Punjab, Haryana, and Uttar Pradesh when farmers switched to an annual rice-wheat cropping system. Farmers began to use combine harvesters in conjunction with high-yielding varieties of rice and wheat crops. One consequence of adopting high yield crops and intensifying farm mechanization was that vast quantities of postharvest crop residue began to be generated in situ [4]. In northern India the high yield rice crop's residue has little use to farmers but by contrast wheat residue is used as cattle fodder. While both crops' residues get burned rice CRB is more prevalent than wheat CRB [25,26]. Each year rice or paddy postharvest residue is burned during October-November and wheat postharvest residue is burned during April-May [21]. The time during which rice residue is burned is a season in northern India when the winter air is relatively stagnant–leading to the formation of heavy smog which does not get easily dispersed (Source: https://weather.com/en-IN/india/science/news/2018-10-30-why-do-pollution-levels-skyrocket-during-winter). In the summer months of April-May, the burning of wheat residue is less evident since the air is not stagnant like in winter. This is plausibly one reason why wheat CRB does not garner as much attention as rice CRB.

Apart from rice residue being of little use to farmers they tend to burn it because of a quick turnaround required for the subsequent wheat crop; the best time to sow wheat runs from November through December–roughly two weeks after rice is harvested [4,27]. The earlier non-high yield varieties of rice and wheat, which are not grown much anymore in northern India, used to have cropping cycles that lasted longer than the currently prevalent high yield varieties that have shorter rotations of 120 days. Farmers would previously produce only one crop–either wheat or rice–in a farming year [13]. With the adoption of short rotation high yield varieties, the yearly trend shifted to rice being grown between June and October and wheat being grown between November and March [4]. With these crop rotations little time is left between the rice harvest in October-November and the sowing of wheat in November-December. Delays in wheat sowing can adversely affect its harvest quality.

The conundrum is that with a gap of 46 days between wheat residue removal and the transplanting of paddy there are no binding constraints on farmers to burn wheat residue [13]. Even with wheat residue being useful as cattle fodder and farmers not being time constrained to sow the subsequent rice crop, there is increasing evidence of wheat residue burning. Studies in the literature have reported on other reasons for CRB in general. These include the seasonal shortages of migrant farm labor [21], the fact that it is seen as a quick disposal method by farmers, and that residue burning is considerably less expensive than its machine removal [4,15,17,27,30,31].

## Data

The primary data for this analysis comes from the district of Karnal in Haryana state, northern India. The district of Karnal is a homogenous agroclimatic zone in northern Haryana and highly productive agriculturally [32]. There are 78,265 farmland holdings in Karnal, covering 388,827 hectares, of which 187,012 hectares are under wheat cultivation [33]. In 2018 we surveyed 1,230 households across 12 villages in Karnal on rice residue management practices [17]. For the current study, we selected a random sub-sample from 8 of the original 12 villages

to be resurveyed in 2019, specifically on their wheat CRB practices. Our target was 40 farmers per village. During the selection process, we excluded farmers who only grew the non-coarse varieties of rice (i.e., basmati) or non-paddy crops since the residues of these crops typically are not burnt. This process resulted in between 35 to 40 farmers for each village for a total of 301 farmers, who grew coarse varieties of rice among other things, participating in the 2019 survey.

The random sampling for this research was designed to answer the questions of whether choosing CRB made economic sense for wheat farmers, and if there is a relationship in the residue management choices between rice and wheat. The 2019 survey took place immediately after the wheat crop harvesting season in the first half of May 2019. Almost all farmers stated that the harvest of the wheat crop takes place during the last week of April and the first week of May. Each survey respondent was either the household head or the principal household person making agricultural decisions.

Each survey collected information on household demographic and socioeconomic information. Additionally, farmers were asked about their wheat residue management practices. The questions included what their options of wheat residue management were, the reasons for choosing to burn wheat residue, their use of residue if not burned, their perceptions on whether residue burning is a common practice in the village, and their perceived effects of residue burning on soil quality and the natural environment. All questionnaires were translated from English to Hindi, see the English version in S1 Appendix.

## Methodology

This paper analyzes crop residue burning (CRB) of wheat residue at three levels. To begin with, we analyze the various factors that influence a farmers' decision to burn wheat residue. We also compare socioeconomic characteristics of households that did and didn't burn wheat residue. Furthermore, to understand if there is a link in the CRB decision between crops, we test for the correlation between households that adopted CRB of wheat with that of rice. In addition to this, we undertake two additional sets of analyses- testing for the benefits and costs of CRB and a regression framework. In order to comply with the Institutional Review Board requirement to interview 301 farmers, we got the study approved by the IRB Board at the American University of Sharjah as per protocol #17-406RN with exemption under 45 CFR 46.101(b)(5). Oral consent was recorded for each participant. Extensive meetings were also held with village-level governing bodies (Panchayat) before the start of survey activities to share information about the study objectives.

### Benefit-cost analysis framework

Postharvest wheat residue can be used as cattle fodder, or alternatively it can be reincorporated into soil, or farmers could also choose to sell it to others. Given that wheat residue can be sold, we estimate a benefit value of not burning wheat residue, per acre. This is estimated as the product of the average residue generated per acre and the average selling price received by farmers. At the same time, we recognize that in order to sell the residue the farmer first has to pay to have it cleared out. Cost estimates for this were collected as part of the survey for both, manual and mechanical removal of the residue. We then compare the benefit value to the cost of removing the residue. If the difference (i.e., benefit–cost) is positive, it indicates that despite having to incur a cost for the removal of the residue it is still beneficial for the farmer to sell it. On the other hand, if the difference is negative (i.e., cost > benefit) then it is in the farmer's interest to not sell the residue, and possibly burn it. We estimate this difference for farmers that did and didn't undertake wheat CRB. This analysis can shed more light on farmers' perceptions of the costs and benefits of burning wheat residue.

## Regression framework

To investigate the factors that influence a farmer's decision to choose CRB we adopt a probit regression framework as given by Eq 1. P(*CRB*) represents the probability that the farmer has burned wheat straw in the last five years, and $X_1, \ldots, X_N$ are independent regression covariates.

$$P(CRB) = \frac{\exp(\beta_0 + \sum_{n=1}^{N} \beta_n X_n)}{1 + \exp(\beta_0 + \sum_{n=1}^{N} \beta_n X_n)}, \tag{1}$$

where $\sum_{n=1}^{N} \beta_n X_n$

$= \beta_1(\text{Paddy CRB}) + \beta_2(\text{Common Practice Wheat CRB}) + \beta_3(\text{Environmental Awareness}) + \beta_4(\text{Diminishing Soil Quality}) + \beta_5(\text{Land Size}) + \beta_6(\text{Number of Machine Owned}) + \beta_7(\text{Education Dummy}) + \beta_8(\text{Livestock Owned}) + \beta_9(\text{Residue Removal Cost}) + \beta_{10}(\text{Residue Price}).$

[34] state the importance of accounting for farmer-level heterogeneity, including demographics and farm characteristics, when characterizing their agricultural practices. Accordingly, in Eq 1 we control for common characteristics that contribute to rice and wheat residue burning as identified previously in [17], namely, farmers' perceptions of burning as a residue management strategy, socioeconomic characteristics such as farmland size (Land size), farm machinery ownership, education (binary), ownership of livestock (binary), and farmer's private benefits (Residue price) and costs (Residue removal cost) associated with residue burning.

Farmer perceptions are measured in terms of whether the respondent believes that wheat residue burning is a common practice in their village (Common Practice CRB), perceptions on the negative environmental effects of CRB (Environmental awareness), and perceptions on the negative effects of CRB on soil quality (Diminishing soil quality). Eq 1 also controls for whether the farmer burned rice residue in the previous season (Paddy CRB). This information is merged with the 2019 data from the original 2018 survey that focused on rice CRB. In 2018, farmers were asked the question "In the last 5 years, was paddy residue ever burned on your land?" (in [17]). There were three possible responses to this question: *Yes*, *No*, and *Unwilling to Answer*. Since some farmers refused to respond to the question about whether or not they undertook rice CRB in the previous season, we estimate Eq 1 in two ways. First, we include a binary for whether or not the farmer responded, and second, we exclude this variable in another specification. The reason for doing this is to infer whether a farmer who is unwilling to answer is equivalent to a farmer who reports burning paddy residue previously.

Eq 1 is also run with and without controlling for a probability weight of household populations in each village. This population weighting method corrects for any representation bias in the sampled villages. An expanded specification of Eq 1 includes two additional covariates on the cost of wheat residue removal per acre using machinery and the price of wheat residue collected per acre. For the latter, we shall exclude those farmers who are not aware of the price of wheat residue. For all specifications, we exclude outlier farmers who have a very large land sizes compared to the average. We list two hypotheses that we can test with our survey data.

**Hypothesis 1**: *If CRB is perceived to be a common practice then an individual farmer will follow suit.*

**Hypothesis 2**: *If a farmer chooses CRB to manage rice residue in one harvest season then the farmer is likely to do the same in managing postharvest wheat residue.*

One could argue about an endogeneity issue arising here, wherein the effect may be reversible or that we face an omitted variable bias when using rice CRB as an explanatory variable. To test for endogeneity, we also consider regressions that include rice residue burning being the common practice (i.e., farmer's perception on whether CRB is a common practice to manage rice residue) as an instrumental variable. The argument for using this as an instrument is that since CRB is perceived to be a common practice in the management of rice residue amongst these farmers, then it presumably affects rice CRB directly and would not directly affect a farmer's wheat CRB decision. This approach would plausibly address the endogeneity issue, if any.

## Results

### Descriptive statistics

In Table 1 we summarize the residue management practices by village. We note that almost all arable land available for wheat cultivation is used to grow either the soft or coarse varieties of wheat. Table 1 shows that many farmers in each village adopt burning as a means of wheat residue removal. The percentages of farmers employing this method in the last five years range from 17.95 to 40.54 percent. Overall, a little less than a third of the respondents reports the burning of wheat residue in the last five years, and about two-thirds deny it. Among those who did not use CRB, more than three-quarters reported they used only machine to remove the residue.

In Table 2, we compare socioeconomics and other aspects among these two groups of farmers–those who undertook wheat CRB (Yes) and those who did not (No). Table 2 shows that the average number of farm assets owned, and the average total cultivated land are significantly

**Table 1. Wheat residue management practices by village.**

| | All | (1) | (2) | (3) | (4) | (5) | (6) | (7) | (8) |
|---|---|---|---|---|---|---|---|---|---|
| Number of respondents | 301 | 39 | 37 | 37 | 35 | 39 | 40 | 37 | 37 |
| Average total cultivated land for wheat (acre) [1] | 7.67 (11.47) | 7.57 (7.25) | 9.60 (9.71) | 14.61 (23.91) | 5.46 (3.85) | 3.14 (2.76) | 4.25 (3.67) | 9.56 (13.38) | 7.63 (7.93) |
| Average cultivated land for soft wheat (acre) [1] | 7.46 (10.20) | 7.58 (7.25) | 9.37 (9.56) | 13.72 (19.62) | 5.41 (3.77) | 3.14 (2.76) | 4.16 (3.58) | 9.13 (12.39) | 7.63 (7.93) |
| Average cultivated land for coarse wheat (acre) [1] | 0.21 (1.90) | 0 | 0.23 (1.06) | 0.89 (4.93) | 0.06 (0.34) | 0 | 0.09 (0.55) | 0.43 (1.85) | 0 |
| Residue ever burned in last 5 years (% yes) | 30.90 | 17.95 | 37.84 | 40.54 | 40.00 | 28.21 | 20.00 | 27.03 | 37.84 |
| Average no. of times burned in last 5 years[1,2] | 2.62 (0.90) | 2.57 (1.13) | 2.71 (1.07) | 2.2 (0.77) | 2.5 (0.94) | 3.09 (0.70) | 2.63 (0.52) | 2.7 (1.06) | 2.71 (0.83) |
| Hire labor for residue removal (% yes)[3] | 9.30 | 5.13 | 2.70 | 2.70 | 5.71 | 15.38 | 22.50 | 5.41 | 13.51 |
| Use machine for residue removal (% yes)[3] | 75.08 | 79.49 | 81.08 | 81.08 | 80.00 | 58.97 | 67.50 | 83.78 | 70.27 |
| Use both machine and labor for residue removal (% yes)[3] | 15.61 | 15.38 | 16.22 | 16.22 | 14.29 | 25.64 | 10.00 | 10.81 | 16.2 |

[1] The numbers show the averages; the standard deviations are shown in parentheses.

[2] This average is for those respondents who state that wheat residue has been burned in their land in the last five years.

[3] The question asks, "When you do not burn wheat residue, which method do you use to remove it?"

**Table 2. Socioeconomic characteristics by wheat residue burning practice.**

| Wheat residue ever burned in last 5 years | Yes | No | (*p*-value)[1] |
|---|---|---|---|
| % of respondents[2] | 30.90 | 67.77 | |
| Burned paddy residue in last 5 years (% yes) [3] | 76.92 | 56.83 | 0.002 |
| Average age of respondents (years) | 42.99 (13.30) | 42.67 (13.11) | 0.847 |
| Average education level of respondents[4] | 3.70 (1.63) | 3.56 (1.64) | 0.495 |
| % of pukka house-holds[5] | 100 | 92.65 | 0.004 |
| Average no. of types of farm asset owned | 1.08 (1.48) | 0.59 (1.20) | 0.003 |
| % own combine harvester | 0.00 | 1.47 | 0.554 |
| Average number of livestock[6] | 4.42 (3.28) | 3.76 (3.78) | 0.148 |
| Average total cultivated land for wheat (acre) | 10.29 (10.36) | 6.37 (11.78) | 0.006 |
| Average cultivated land for soft wheat (acre) | 10.13 (10.12) | 6.15 (10.01) | 0.002 |
| Average cultivated land for coarse wheat (acre) | 0.16 (1.12) | 0.23 (2.18) | 0.788 |
| Average total cultivated land for paddy (acre) | 11.32 (11.10) | 6.85 (12.20) | 0.003 |
| Residue removal method (if not burned)<br>• labor removal<br>• machine removal<br>• mixed of both | 6.45<br>84.95<br>8.60 | 10.78<br>70.10<br>19.12 | 0.021 |
| Average cost of manual residue removal (rupees per acre) | 3190.32 (325.73) | 3136.68 (341.88) | 0.206 |
| Average price of wheat residue collected (rupees per acre) | 4704.30 (1738.18) | 5186.9 (3424.03) | 0.219 |
| *Farmer perceptions on three separate aspects*: | | | |
| 1. Wheat residue burning is common in the village (% yes)[7] | 73.12 | 16.18 | 0.000 |
| 2. Aware of negative environmental effect (% yes) | 79.57 | 94.61 | 0.000 |
| 3. Effect of burning on soil quality (% yes):<br>• improves quality<br>• diminishes quality<br>• No effect/don't know | 5.38<br>23.66<br>70.97 | 0.49<br>65.69<br>33.82 | 0.000 |

[1] We use a two-tailed t-test to test the equality of means (standard deviations are shown in parentheses) and Fisher's Exact test to test the equality of proportions.

[2] We exclude those respondents who were unwilling to answer chosen questions.

[3] From the 2018 survey on paddy residue management.

[4] 0 –No education/illiterate; 1 –semiliterate (never attended school); 2 –Primary (Grades 1–5); 3 –Middle (Grades 6–-8); 4 –Secondary (Grades 9–10); 5 –Higher Secondary (Grades 11–12); 6 –Graduate; 7 –Postgraduate.

[5] In descending order of quality, housing construction materials are pukka (brick and mortar), semi-pukka (mix of brick, mortar, and clay), and kuccha (clay).

[6] Includes only larger livestock—buffaloes, cows and bullocks.

[7] The question asked is "Is burning wheat residue a common practice in your village?" This is question number 2.2 on the questionnaire in the Appendix. Options for answers are: 1 = not at all, 2 = occasionally, 3 = often, 4 = very common, 5 = everybody does it. The reported number here is the percentage of two or higher values.

higher amongst farmers who burn wheat residue than those who reportedly do not burn it. Since farm assets and land are proxies for farmer wealth, this implies that those who burn wheat residue are presumably wealthier.

Table 2 also shows that a greater proportion of farmers who use CRB perceived CRB as a common practice than those who did not use CRB. This suggests that farmers are more likely to use CRB if they consider it to be a common practice in their village. In contrast, a greater proportion of farmers who did not burn wheat were aware of CRB's negative implications for soil health and environmental outcomes. That is, the awareness of the negative environmental effects of burning and the perception of its adverse effect on soil quality appears to discourage farmers from burning residue. In Table 2, we also see that the average cost of manually removing the wheat residue and the average market value of the collected residue are not significantly

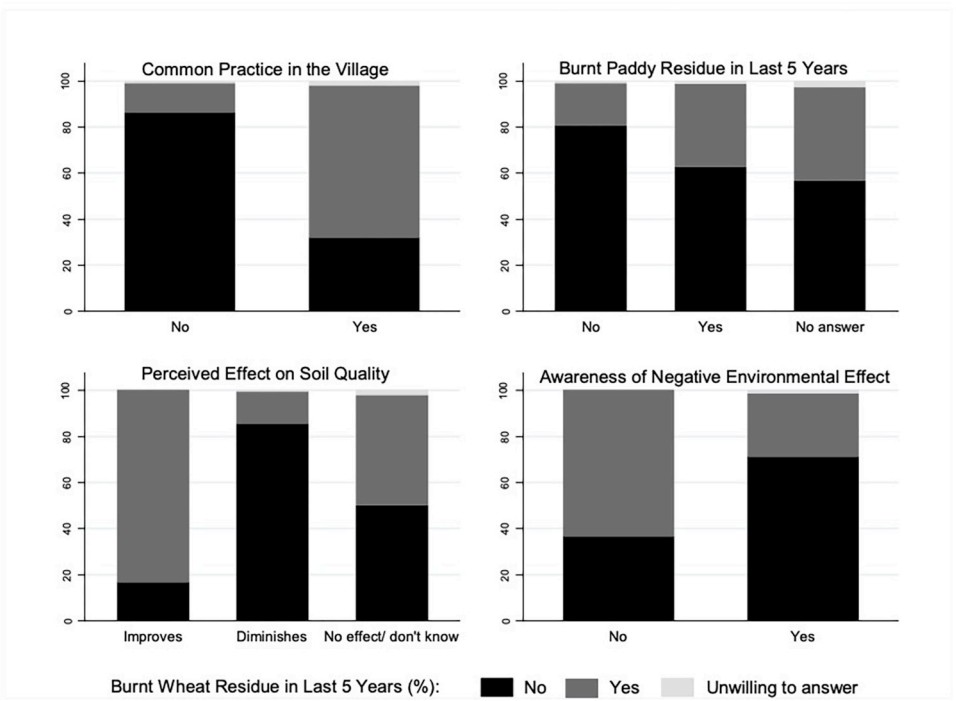

**Fig 1. Percentage of wheat residue burning based on farmer perceptions and paddy residue burning.**

different between the farmers who burn wheat residue and the ones who do not. However, the two groups relied on different methods of residue removal with the 'yes' group relied more on mechanical removal of residue while the 'no' group relied relatively more on manual removal.

## Linking CRB for wheat with that for rice

Table 2 also indicates a significant correlation between the two decisions of burning wheat residue and rice residue in recent years. Nearly 80% of the subset of farmers that undertook CRB for wheat also undertook CRB of rice in the previous season. In contrast less than 60% of farmers in the 'No' group burned rice residue in the previous season.

These findings related to CRB of rice and the effects of farmer's perceptions related to the common practice in the village, the effect of residue burning on soil quality, and the environmental impact of their wheat residue burning decision are also mirrored in Fig 1. The top right panel of Fig 1 presents the distribution of farmers in terms of their wheat residue burning behavior as related to their rice residue burning decisions over the last five years. It clearly indicates that farmers who did not burn paddy residue in the last five years are less likely to burn wheat residue as opposed to those who did burn paddy residue in the last five years or

**Table 3. Primary reasons for choosing CRB.**

| *Which factor in the decision to burn crop residue is most important to you?* | | |
|---|---|---|
| | **Wheat (2019)** | **Paddy (2018)** |
| Combine harvester | 77.08% | 16.61% |
| Expensive alternative options | 17.94% | 57.81% |
| No other use | 2.66% | 12.96% |
| Others | 2.33% | 12.30% |

those who were unwilling to answer this question. This consistency of burning behavior across two crops may arise from underlying factors that similarly affect both decisions. We explore this further in the regression results section.

Although we see that burning of rice residue in one season is correlated with CRB for wheat in the following season, it appears that the reasons for CRB of the two crops are not the same (Table 3). While nearly 60% of farmers cited cost-effectiveness as the most important reason for choosing CRB for rice residue management, less than one-fifth of farmers considered it to be so when it comes to CRB for wheat. For the latter, 77.08 percent of respondents claim that the most important reason for choosing CRB is the use of a combine harvester. As alluded to in the section on Background of CRB in India, combine harvesters were a part of the intensive farm mechanization process since the 1970s, which consequently led to vast quantities of post-harvest crop residue being generated. The high cost of managing wheat residue without burning it is the second-most important reason. This is akin to findings in [20] wherein the upfront cost of residue harvesting machinery is an important barrier to adopting environmentally friendly alternatives among Indian rice and wheat farmers. Similarly, [19] report that limitations to farmer's resources and information hinder the adoption of cleaner production strategies. The lack of residue usability is seen as a reason for choosing CRB by 12.96 percent of paddy farmers as opposed to a smaller 2.66 percent amongst wheat farmers.

## Benefit-cost analysis of wheat residue burning

Using the primary data collected from 301 wheat farmers in Haryana, we infer that more than 90 percent of farmers report that wheat residue is commonly used as cattle fodder. Others report that they reincorporate wheat residue into soil and/or sell the residue. On average, farmers report that wheat generates 14.87 quintals of residue per acre, and the selling price is 332.54 Indian Rupees per quintal. This yields an estimate of the gross benefit of wheat residue at INR4,945 or US$70.64 per acre (INR70 = US$1 in April 2019). The survey data show an average cost of hiring labor to manually remove postharvest wheat residue as INR3,153/acre (US$45.04/acre) of farmland. Moreover, our data indicate that 75 percent of farmers use machinery to remove wheat residue; they report an average cost of INR1,642/acre (US$23.46/acre). Removal of the wheat residue using machinery is less costly than using manual labor. Moreover, we observe that the differences between the benefit and cost (net benefit) of using residue removal machinery are positive for all participants. We further note that the average net benefits for participants who did not burn and those who burned their wheat residue are INR3,317 ($47.38) and INR3,030 ($43.28) per acre, respectively. However, this difference is not statistically significant ($p$-value = 0.21 for a two-sided $t$-test). These numbers suggest that there are other reasons preventing farmers from using a non-CRB method.

Most farmers prefer to sell their excess wheat residue to neighboring farmers who need it as cattle fodder or reincorporate it into the soil. As indicated by these prices and costs, selling wheat residue in Karnal district's fodder markets would make greater economic sense for a farmer if they had collected residue using machinery instead of manually. They would gain US$25.6 per acre (i.e., $70.64 - $45.04) on average by manually collecting wheat residue and selling it. However, if the residue was collected using machinery, that would yield a higher average net benefit of US$47.18 per acre (i.e., $70.64 - $23.46). These estimates can be compared to the per acre cost of manually removing rice residue calculated in our previous study [17], which yielded an estimated average net cost of US$51 per acre of rice residue removal. These results are corroborated by Table 1, in which we noted that three-quarters of the farmers use machinery to remove wheat residue while less than a tenth of them use manual labor.

While wheat farmers could potentially gain from selling their surplus wheat residue in fodder markets, the limiting factors remain inadequate access to residue markets in the district and possibly high market transportation costs. Even though wheat residue can be traded in government fodder markets [35], there exists considerable variability in its price and uncertainty about its salability [36]. Besides, given that wheat residue burning remains widespread [22], this suggests that at present the district fodder markets are possibly not offering cost-effective alternatives to CRB. This conundrum of burning wheat residue despite the positive net benefit of selling it shall be explored further in the next section.

## Regression results

Table 4 reports five probit model results using Eq 1 as described in the Methodology section. In Models 1 and 3 we exclude 36 farmers who were not willing to answer whether they have used CRB after harvesting paddy. We include them in Models 2 and 4 with a binary variable indicating whether they were willing to answer or not. We exclude two outliers with land size greater than 50 acres. Models 3–5 include the probability weights of household populations in

**Table 4. Regression coefficients of determinants of wheat residue burning using probit.** (Binary dependent variable: choosing CRB in last five years–Yes = 1, No = 0).

| Variable | (1) | (2) | (3) | (4) | (5) |
|---|---|---|---|---|---|
| Paddy Residue Burning (Yes) | 0.60 (0.38) | 0.63* (0.38) | 0.95* (0.50) | 0.97* (0.52) | 0.93* (0.56) |
| Paddy Residue Burning (No Answer) | | 0.87*** (0.23) | | 0.83*** (0.21) | 0.76*** (0.25) |
| Common Practice | 1.54*** (0.30) | 1.51*** (0.32) | 1.23*** (0.37) | 1.22*** (0.32) | 1.18*** (0.25) |
| Environmental Awareness | -0.45 (0.30) | -0.48 (0.27) | -0.31 (0.35) | -0.33 (0.33) | -0.38 (0.47) |
| Soil Quality (Diminish) | -1.17*** (0.33) | -1.11*** (0.22) | -1.14*** (0.31) | -1.11*** (0.24) | -1.11 (0.24) |
| Land Size | 0.04 (0.03) | 0.03 (0.02) | 0.03 (0.03) | 0.03 (0.02) | 0.03 (0.02) |
| Number of Machines Owned | 0.08 (0.11) | 0.06 (0.09) | 0.07 (0.12) | 0.02 (0.11) | 0.01 (0.12) |
| Education | 0.46** (0.21) | 0.37* (0.22) | 0.29 (0.21) | 0.21 (0.24) | 0.15 (0.25) |
| Livestock Owned (Dummy) | -0.08 (0.38) | -0.15 (0.24) | -0.36 (0.31) | -0.25 (0.26) | -0.20 (0.29) |
| Cost of Machine Residue Removal per acre ('000 INR) | | | | | 0.04 (0.13) |
| Price of Wheat Residue Collected per acre ('000 INR) | | | | | 0.02 (0.04) |
| Population Weights | No | No | Yes | Yes | Yes |
| Pseudo R-Squared | 0.45 | 0.43 | 0.41 | 0.40 | 0.40 |
| Number of Observations | 259 | 295 | 259 | 295 | 262 |

1. *, **, *** indicate error levels of 10%, 5%, and 1%, respectively.

2. Robust standard errors clustered by village are in parentheses.

3. All models include a constant.

4. We exclude four subjects who were not willing to answer whether they had burned wheat residue in the last five years. Interpreting this answer as "yes" provides similar results.

5. We exclude 2 outliers (land size greater than 50 acres); changing to land size greater than 40 or 60 acres does not alter our main results.

6. Refer to Table 2 for variable definitions.

each village. The number of wheat-growing households are sourced from the village panchayat lists. Model 5 includes two additional covariates. We exclude 35 farmers who report that they do not know the price of wheat residue. All the five models yield remarkably similar results. We discuss seven salient findings.

We find firstly that our results support Hypothesis 1; i.e., farmers who believe that CRB is a common practice are more likely to choose CRB than those who do not believe so (*p*-value < 0.01 in all models). Second, the perception of the effects of CRB on soil quality has a significant impact on farmer behavior. Farmers who think that CRB diminishes soil quality are significantly less likely to choose CRB than those who think it improves soil quality or are uncertain about its effect. This result suggests that farmers do internalize their associated costs and benefits of CRB. Third, farmer's awareness of the negative environmental impact of CRB has a negative sign as one would expect; the effect however is statistically insignificant. Thus, farmers do not substantially internalize the external social costs of CRB in their private decisions. These three findings are consistent with our previous findings amongst a larger set of farmers with regards to their rice residue management practices [17].

Fourth, amongst the various socioeconomic characteristics, education is the only variable in which we observe any statistical significance. Farmers with higher education are more likely to adopt CRB (*p*-value = 0.029 for Model 1, 0.093 for Model 2, and not significant in Models 3 and 4). The coefficient of land size is positive as expected but statistically insignificant. [17] report that paddy farmers with larger land sizes are significantly more likely to use CRB. The coefficient of the number of machines owned is also positive but not statistically significant since only 3 farmers reported owning a harvester. Livestock ownership has a negative and insignificant coefficient.

Fifth, incorporating a probability weight of household populations in each village–as done in Models 3 and 4 –results in qualitatively similar findings to Models 1 and 2, which exclude these weights. This result indicates a zero bias of population representation in the regression results.

Sixth, for those farmers who reported earlier that they burned rice residue, and even more so for those farmers who were unwilling to answer this question, we note that both groups significantly more likely to burn wheat residue as compared to farmers who did not burn rice residue previously. This result supports Hypothesis 2 –that CRB decisions for rice and wheat are potentially linked, or that there is a spillover effect from rice CRB to wheat CRB.

The test for endogeneity in our regression framework when using rice CRB as an explanatory variable relies on the *p*-values from the Wald-test. Using rice residue burning being a common practice as an instrument yields the *p*-values from the Wald-test of exogeneity as 0.32 and 0.57 –when including and excluding probability weights of household populations, respectively. These non-significant p-values imply that we cannot reject the null hypothesis of exogeneity. This suggests that not including an instrumental variable in the regression framework is reasonable. These instrumental variable regression results are available upon request.

Lastly, the coefficients of the cost of wheat residue removal using machinery per acre and the price of wheat residue collected per acre in Model 5 are not statistically significant. We also replace the cost of residue removal using machinery with labor as well as use the net benefit (the difference between the price of residue collected and the cost of residue removal per acre) to replace both variables; none of these coefficients is statistically significant. We noted from the benefit-cost analysis that a high cost of transporting crop residue to fodder markets limits farmers from adopting environmentally friendly ways of residue management.

In Table 5, we report the average marginal effects from the probit regressions results of Models 3, 4, and 5. The results of all the models are similar, and there are three variables where their average marginal effects are statistically significant. First, farmers who believe that CRB is

**Table 5. Average marginal effects on wheat residue burning (using Models 3, 4, and 5).**

| Independent Variable | Marginal Effects | | |
|---|---|---|---|
| | Model (3) | Model (4) | Model (5) |
| Paddy Residue Burning (Yes) | 0.194* (0.107) | 0.210* (0.119) | 0.207 (0.130) |
| Paddy Residue Burning (No Answer) | | 0.175*** (0.059) | 0.165** (0.067) |
| Common Practice | 0.251*** (0.063) | 0.261*** (0.055) | 0.261*** (0.046) |
| Environmental Awareness | -0.063 (0.074) | -0.071 (0.068) | -0.085 (0.102) |
| Soil Quality (Diminish) | -0.233*** (0.058) | -0.238*** (0.052) | -0.245*** (0.053) |
| Land Size | 0.007 (0.005) | 0.007 (0.005) | 0.008 (0.005) |
| Number of Machines Owned | 0.013 (0.024) | 0.003 (0.025) | 0.003 (0.026) |
| Education | 0.060 (0.044) | 0.043 (0.050) | 0.033 (0.057) |
| Animals Owned (Dummy) | -0.073 (0.068) | -0.052 (0.058) | -0.044 (0.067) |
| Cost of Machine Residue Removal | | | 0.005 (0.040) |
| Price of Wheat Residue Collected | | | 0.008 (0.016) |

1. *, **, *** indicate error levels of 10%, 5%, and 1%, respectively.

2. Robust standard errors clustered by village are in parentheses.

a common practice are 25 to 26 percent more likely to choose CRB than do those who do not believe so ($p$-value $< 0.001$). Second, farmers who believe that using CRB diminishes soil quality are 23 to 26 percent less likely to choose it as compared to other farmers who believe otherwise ($p$-value $< 0.001$). Finally, those farmers who previously chose CRB for rice residue management, and those farmers who were not willing to reveal their CRB choices, are 19.4 to 21 percent and 15.6 to 17.5 percent more likely, respectively, to choose CRB for wheat than farmers who did not previously employ CRB for rice. The average marginal effects of other control variables–socioeconomic characteristics, benefit of selling residue, and cost of manual residue removal–are not statistically significant; suggesting that these factors are not major determinants of wheat CRB.

## Discussion

This study has taken a closer look at the reasons behind crop residue burning (CRB) of wheat. Consistent with our previous findings concerning rice CRB [17], we infer that the perception of residue burning being a common practice significantly increases wheat CRB, while the perception that residue burning lowers soil quality makes the practice less likely. We similarly find no evidence of environmental awareness lowering residue burning, thereby indicating a public goods characteristic. On tracking the residue management choices of rice-wheat farmers over two consecutive cropping seasons, the current study infers that 31 percent of farmers burn their wheat residue and that they are 21 percent more likely to do so if they had previously burned rice residue. We also determine that 92 percent of the farmers use some portion of wheat residue as cattle fodder, 52 percent reincorporate some of it into the soil, and 39 percent choose to sell some portion of it.

The benefit-cost analysis suggests that wheat residue attains a potentially positive value net of residue collection costs–more so for machine collection of residues rather than manual collection. While farmers in Haryana are known to use wheat residue as fodder, it is seen to be the case that several of them burn it even when there is little compulsion to do so. This paper has evaluated evidence relating to wheat CRB occurring notwithstanding a positive net benefit of selling it. This counterintuitive result has been examined in the context of rice CRB decisions possibly influencing the same farmers' choices concerning wheat CRB in a subsequent

cropping season. Farmers appear to experience a spillover from their previous rice residue burning practices even though it may not be their most rational choice. This suggests that policies designed to curb the use of CRB as a farm management strategy should not focus on specific crops in isolation. Rather, it is critical to inform the design of such policies by looking at the agricultural calendar as a whole and realizing that farmers' decisions related to the management of one crop can have spillover effects for a second crop.

Meeting the challenge of crop residue burning requires carefully targeted policy interventions to work effectively, thereby making it imperative that policy interventions are designed in accordance with how people behave [37,38]. Accordingly, the identification of pertinent behavioral factors driving crop residue burning could bolster the design of effective policies to reduce its occurrence. Wheat CRB was higher when farmers perceived it to be a common method of managing crop residue. In this respect, policymakers could use informational campaigns or nudges to encourage individuals to conform to social norms of pro-environmental behaviors. For instance, [39] provide a meta-analysis of field experiments that rely on social norms in promoting pro-environmental choices; [18] consider the same in a crop residue burning context.

Farmers could potentially gain from selling their surplus wheat residue. However, limiting factors remain for the farmers–such as an inadequate access to residue markets, uncertainty about the marketability of fodder, variability in fodder prices, and high market transportation costs. Ensuring that farmers have access to well- functioning markets for wheat residue could avert the scaling-up of wheat residue burning amongst Indian wheat growers. Such markets could prove financially remunerative to farmers by increasing their perceived net benefit of pro-environmental practices. An avenue worth exploring here might be to subsidize the cost of transporting residue to markets. The persistence of pro-environmental choices could arise with assured financial gain for the farmers. [16] report that the economic benefits of sustainable alternative uses of crop residue should be made clear to farmers for policies to effectively reduce CRB. In a similar vein, [40] finds that farmers persist with environmentally-friendly tillage decisions based even on temporary financial incentives.

## Conclusion

The burning of postharvest crop residue (CRB) has come to acquire significance from the perspectives of sustainable farmland management practices, the natural environment, and public health in developing economies with large agrarian sectors. Addressing the drivers of farm management practices by agricultural stakeholders assumes relevance from the standpoint of sustainable land use. India, with its large agrarian sector, regularly witnesses postharvest crop residue burning and bears substantial public health and environmental costs. While rice residue burning worsens air quality every winter and continues to receive a lion's share of the attention in the literature, wheat residue burning on the other hand, although known to occur every summer, has not received much attention. This paper addresses this gap by examining the drivers of wheat residue burning and drawing a relationship to CRB of rice.

With scarce attention paid to the phenomenon of wheat residue burning in India, our paper is the first to investigate its drivers. One reason identified for residue burning is the intensive mechanization of agriculture since the 1970s such as the adoption of combine harvesters for the rice-wheat dual cropping system in northern India. While rice residue burning occurs regularly on a wide scale every winter, the novelty of this paper is that it turns the lens on the possibility that farmers shadow their rice residue management practices from prior seasons when it comes to the wheat harvest.

We note that we can only cautiously extrapolate or generalize our findings to the broader region of this part of India since this study is limited by the size of its sampled farmers and the choice of a single district in the state of Haryana. Furthermore, our data pertains to a single crop harvest season, so we cannot discern long-term trends, for which the sampled households would need to be repeatedly surveyed on their CRB decisions over successive seasons. Despite the data-driven limitations, this study does identify an emerging yet important gap in the knowledge of CRB occurrences. Crop residue burning is topical to several developing countries and this paper contributes to a growing narrative on agricultural residue management practices that have wider environmental and public health consequences. Future research calls for the conceptualization of targeted policies that foster pro-environmental agricultural practices.

## Supporting information

**S1 Appendix. Survey questionnaire.**
(DOCX)

**S1 File. Raw data table.**
(XLSX)

## Acknowledgments

We acknowledge the receipt of a faculty research grant (#FRG17-R-49) and the Open Access Program (#OAPSBA-1210-B00016) from the American University of Sharjah. This paper represents the opinions of the authors and does not mean to represent the position or opinions of the American University of Sharjah. We thank Probe and Search India for their professional survey work and Soumya Gupta for her comments on the initial draft of this paper. We also thank seminar participants at the American University of Sharjah and participants at the Southern Economics Association Meetings in 2022 for their comments and suggestions.

## Author Contributions

**Conceptualization:** Adrian A. Lopes, Dina Tasneem, Ajalavat Viriyavipart.

**Data curation:** Dina Tasneem.

**Formal analysis:** Adrian A. Lopes, Ajalavat Viriyavipart.

**Funding acquisition:** Adrian A. Lopes, Ajalavat Viriyavipart.

**Investigation:** Adrian A. Lopes, Dina Tasneem, Ajalavat Viriyavipart.

**Methodology:** Adrian A. Lopes, Dina Tasneem, Ajalavat Viriyavipart.

**Project administration:** Adrian A. Lopes.

**Visualization:** Dina Tasneem.

**Writing – original draft:** Adrian A. Lopes, Dina Tasneem, Ajalavat Viriyavipart.

**Writing – review & editing:** Adrian A. Lopes, Dina Tasneem, Ajalavat Viriyavipart.

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
