## [Decision Letter · Decision Letter 0]

19 Jul 2023

PONE-D-23-10161Determinants of Wheat Residue Burning: Evidence from IndiaPLOS ONE

Dear Dr. Ajalavat Viriyavipart,

Thank you for submitting your manuscript to PLOS ONE. After careful consideration, we feel that it has merit but does not fully meet PLOS ONE’s publication criteria as it currently stands. Therefore, we invite you to submit a revised version of the manuscript that addresses the points raised during the review process.

We look forward to receiving your revised manuscript.

Kind regards,

Sudeshna Bhattacharjya, Ph.D

Academic Editor

PLOS ONE

Journal Requirements:

We will update your Data Availability statement to reflect the information you provide in your cover letter."

6. We note that Figure 1 in your submission contain map/satellite image which may be copyrighted. All PLOS content is published under the Creative Commons Attribution License (CC BY 4.0), which means that the manuscript, images, and Supporting Information files will be freely available online, and any third party is permitted to access, download, copy, distribute, and use these materials in any way, even commercially, with proper attribution. For these reasons, we cannot publish previously copyrighted maps or satellite images created using proprietary data, such as Google software (Google Maps, Street View, and Earth). For more information, see our copyright guidelines: http://journals.plos.org/plosone/s/licenses-and-copyright.

Reviewers' comments:

Reviewer's Responses to Questions

**Comments to the Author**

1. Is the manuscript technically sound, and do the data support the conclusions?

Reviewer #1: Yes

Reviewer #2: Partly

Reviewer #3: Yes

2. Has the statistical analysis been performed appropriately and rigorously? 

Reviewer #1: Yes

Reviewer #2: Yes

Reviewer #3: Yes

3. Have the authors made all data underlying the findings in their manuscript fully available?

Reviewer #1: Yes

Reviewer #2: Yes

Reviewer #3: Yes

4. Is the manuscript presented in an intelligible fashion and written in standard English?

Reviewer #1: Yes

Reviewer #2: Yes

Reviewer #3: Yes

5. Review Comments to the Author

Reviewer #1: Manuscript ID: PONE-D-23-10161

Journal Name: PLOS ONE

Manuscript title: Determinants of Wheat Residue Burning: Evidence from India

The manuscript is based on survey data collected over two years from India and analysis of the surveyed data. The work has been done on a noble topic which is less documented in the literature. The findings could be useful for managing the crop residue burning of wheat that could be a major problem in future. The manuscript is well-written and I have enjoyed reading the article. However, a few things need to be amended before publication.

1. The introduction seems to be lengthy and needs to be shortened. A few paragraphs more resemble “results and discussion” and need to be placed appropriately.

2. The data collection has been done by a service provider and authors are not directly involved in the data collection process. Hence, I am eager to know the validity of the collected data or the authenticity of the data. Was it validated by the authors, if yes how?

3. A paragraph describing the limitations of the study needs to be added.

4. The conclusion seems very lengthy and contains many references. Conclusions should be concise and based on the understanding of the authors on results and discussions. It now looks like the discussion with cited references.

5. The article needs to be checked thoroughly for writing, punctuation marks, etc.

The article may be accepted after “Major revision”.

Reviewer #2: The introduction and background section of the paper need to be refined further to add more clarity in the 'statement of problem'. Also, include a few more studies especially on residue management/CRB practices of rice in the western as well as southern regions of India. Explain the research design and sampling pattern briefly in the methodology. Results of the study are promising though the sample size is small. Results and discussion of 3.2 could be highlighted more with some new insights from the study in place of what is known already. Conclusion needs a major revision. Be more precise in presenting the inferences of the study rather than giving a summary of the work in conclusion. Try not to include any references in this session.

Reviewer #3: The data in the manuscript support the conclusions drawn. The experiment seems rigorous enough; whereas, the results of perceived effect of residue burning on soil quality and awareness of farmers on negative environmental effects seem to have not been captured properly.

6. PLOS authors have the option to publish the peer review history of their article (what does this mean?). If published, this will include your full peer review and any attached files.

Reviewer #1: **Yes: **Surajit Mondal, PhD

Reviewer #2: No

Reviewer #3: **Yes: **Ashish Santosh Murai

---

## [Author Response · Author response to Decision Letter 0]

11 Oct 2023

Please see the attachment 'Reviewer Comments and Responses'.

---

## [Decision Letter · Decision Letter 1]

6 Dec 2023

Determinants of Wheat Residue Burning: Evidence from India

PONE-D-23-10161R1

Dear Dr. Ajalavat Viriyavipart,

We’re pleased to inform you that your manuscript has been judged scientifically suitable for publication and will be formally accepted for publication once it meets all outstanding technical requirements.

Kind regards,

Sudeshna Bhattacharjya, Ph.D

Academic Editor

PLOS ONE

Additional Editor Comments (optional):

Reviewers' comments:

Reviewer's Responses to Questions

**Comments to the Author**

1. If the authors have adequately addressed your comments raised in a previous round of review and you feel that this manuscript is now acceptable for publication, you may indicate that here to bypass the “Comments to the Author” section, enter your conflict of interest statement in the “Confidential to Editor” section, and submit your "Accept" recommendation.

Reviewer #2: All comments have been addressed

2. Is the manuscript technically sound, and do the data support the conclusions?

Reviewer #2: Yes

3. Has the statistical analysis been performed appropriately and rigorously? 

Reviewer #2: Yes

4. Have the authors made all data underlying the findings in their manuscript fully available?

Reviewer #2: Yes

5. Is the manuscript presented in an intelligible fashion and written in standard English?

Reviewer #2: Yes

6. Review Comments to the Author

Reviewer #2: The introduction, methodology, results and conclusion are well written. Still, there is scope for improving the discussion.

7. PLOS authors have the option to publish the peer review history of their article (what does this mean?). If published, this will include your full peer review and any attached files.

Reviewer #2: No

---

## [Editor Report · Acceptance letter]

19 Dec 2023

PONE-D-23-10161R1 

PLOS ONE

Dear Dr. Viriyavipart, 

I'm pleased to inform you that your manuscript has been deemed suitable for publication in PLOS ONE. Congratulations! Your manuscript is now being handed over to our production team.

Kind regards, 

on behalf of

Dr. Sudeshna Bhattacharjya 

Academic Editor

PLOS ONE